# Implementation of Sample Pooling Procedure Using a Rapid SARS-CoV-2 Diagnostic Real-Time PCR Test Performed Prior to Hospital Admission of People with Intellectual Disabilities

**DOI:** 10.3390/ijerph18179317

**Published:** 2021-09-03

**Authors:** Antonino Musumeci, Mirella Vinci, Francesca L’Episcopo, Alda Ragalmuto, Vincenzo Neri, Michele Roccella, Giuseppe Quatrosi, Luigi Vetri, Francesco Calì

**Affiliations:** 1Oasi Research Institute-IRCCS, Via Conte Ruggero 73, 94018 Troina, Italy; amusumeci@oasi.en.it (A.M.); flepiscopo@oasi.en.it (F.L.); aragalmuto@oasi.en.it (A.R.); vneri@oasi.en.it (V.N.); cali@oasi.en.it (F.C.); 2Department of Psychology, Educational Science and Human Movement, University of Palermo, 90128 Palermo, Italy; michele.roccella@unipa.it; 3Department of Sciences for Health Promotion and Mother and Child Care “G. D’Alessandro”, University of Palermo, 90128 Palermo, Italy; giuseppe.quatrosi01@community.unipa.it

**Keywords:** RT-PCR, SARS-CoV-2, sample pooling, sensitivity, hospital admission

## Abstract

Reliability, accuracy, and timeliness of diagnostic testing for SARS-CoV-2 infection have allowed adequate public health management of the disease, thus notably helping the timely mapping of viral spread within the community. Furthermore, the most vulnerable populations, such as people with intellectual disability and dementia, represent a high-risk group across multiple dimensions, including a higher prevalence of pre-existing conditions, lower health maintenance, and a propensity for rapid community spread. This led to an urgent need for reliable in-house rapid testing to be performed prior to hospital admission. In the present study, we describe a pooling procedure in which oropharyngeal and nasopharyngeal swabs for SARS-CoV-2 detection (performed prior to hospital admission using rapid RT-PCR assay) are pooled together at the time of sample collection. Sample pooling (groups of 2–4 samples per tube) allowed us to significantly reduce response times, consumables, and personnel costs while maintaining the same test sensitivity.

## 1. Introduction

Severe acute respiratory syndrome coronavirus 2 (SARS-CoV-2) is a highly transmissible and pathogenic disease caused by a novel RNA virus belonging to the same family as SARS-CoV and Middle East respiratory syndrome coronavirus (MERS-CoV) [1]. SARS-CoV-2 was firstly identified as the cause of an epidemic viral pneumonia in January 2020, with its first outbreak in the Chinese city of Wuhan. From Wuhan, the pandemic rapidly spread across the globe, causing more than 142 million infections and over 3 million deaths (April 2021) [2].

The clinical spectrum of coronavirus disease is highly variable (ranging from asymptomatic/paucisymptomatic, mild or moderate symptoms, severe/critical stage, to death). It is vital to rapidly and reliably identify and isolate cases to prevent possible outbreaks of contagion. It is, therefore, essential to detect virus carriers, along with all their contacts (family members, colleagues, etc.) in order to effectively contest and eradicate viral spread. Furthermore, it is particularly important for testing to be widely available to at-risk communities, which are greatly impacted by the current pandemic. Nearly all hospital (pre)admissions are now consistently preceded by SARS-CoV-2 testing with the aim of protecting the community from the risk of infection and avoiding the emergence of real outbreaks. However, not all diagnostic tests have been proven to have the same sensitivity and reliability [3].

At the moment, the best rapid and reliable diagnostic procedure is the oropharyngeal and nasopharyngeal RT-PCR swab test. Costs and times are quite variable depending on the different types of tests used: rapid, antigen, molecular, serological, salivary, and tear tests [4]. Optimizing human and economic resources, as well response times, will allow for profitable results. The use of pooled tests perfectly meets the optimization criteria as it allows the simultaneous analysis of different biological samples; subsequent individual tests are then performed in case of positivity for SARS-CoV-2.

In our study, we tested the sensitivity of the Credo Diagnostics Biomedical VitaPCR platform using multiple SARS-CoV-2 positive samples analyzed individually and then pooled with one or more negative samples. The pooling strategy allowed us to significantly reduce response times, single exam costs, and personnel costs while maintaining the same test sensitivity standards.

## 2. Materials and Methods

The data in this study refer to a 47-day period from 28 September 2020 to 3 December 2020. This study was approved by the local ethics committee “ComitatoEtico IRCCS Sicilia-Oasi Maria SS” (Prot. CE/149—5 May 2020).

The Oasi Research Institute is a 352-bed research and clinical hospital dealing with the prevention, diagnosis, treatment, and rehabilitation of people with neurodevelopmental and neurocognitive disorders. Following the outbreak of COVID-19, before admission, both patients and accompanying persons are subjected to in-house oropharyngeal and nasopharyngeal swabs for SARS-CoV-2 detection. Moreover, medical interviews and examinations are consistently performed to detect any possible signs and symptoms compatible with COVID-19. If a patient should present with fever or respiratory symptoms during hospitalization, oropharyngeal and nasopharyngeal swabs are carried out again. Finally, with a view to reinforcing prevention, Oasi’s health professionals are subjected to monthly checks. 

Three VitaPCR SARS-CoV-2 platforms (Menarini Diagnostics and Credo Diagnostics Biomedical) were used to carry out the tests. The in vitro rapid molecular diagnostic test was used (reverse transcription polymerase chain reaction (RT-PCR) technology) for the semiquantitative determination of viral RNA in oropharyngeal and nasopharyngeal swabs.

All the subjects who tested positive for SARS-CoV-2 were further resubjected to RT-PCR analysis with different instruments at the SARS-CoV-2 regional reference centers.

VitaPCR platform: The two SARS-CoV-2 RNA target sequences are found in the (specific) virus nucleocapsid (N) gene region and in the conserved region of SARS-like viruses (which includes the SARS-CoV-2, SARS-CoV, and SARS-like bat coronaviruses), respectively. There is also an additional probe to detect human beta-globin (HBB), used as sample adequacy control (SAC), to ensure the correct amount of sample taken and to monitor any inhibition factors in the PCR process. The test involves the use of a buffer (alkaline) in which the oropharyngeal and nasopharyngeal swabs taken from the patient are placed. The reaction (30 μL of the buffer in a lyophilized reagent) is loaded into the VitaPCR platform (Figure 1).

The response time for sample analysis is approximately 20 min. The daily use of point-of-care instruments has allowed the analysis of 3 swabs/hour. VitaPCR SARS-CoV-2 assay detects 3.0 copies/μL of RNA (cDNA) transcripts of SARS-CoV-2 with ≥95% confidence level.

Pooled test: We propose a testing strategy for when large numbers of people need to be screened. We introduced the pooling test of samples before RT-PCR amplification, and the analysis of single samples was carried out only in the case of positive results, thus reducing the number of tests needed. Indeed, biological samples (oropharyngeal and nasopharyngeal swabs) for the SARS-CoV-2 test of 20 positive patients were analyzed individually, and subsequently, in the same reaction buffer, they were aggregated with the swab of a negative sample and then reanalyzed in the absence of dilution effects. The mean of the Ct values and the relative standard deviation (SD) of the two experiments (single and pooled) were calculated for each pair of SARS-CoV-2 and SARS-CoV probes of the test, in all 20 subjects.

Once the new protocol was tested, we performed the test by aggregating up to 4 swabs from different subjects. In particular, the single analysis was performed in samples of patients without an accompanying person or in subjects with COVID-19 symptoms. The pooling analysis of 2 samples was carried out mainly in patients with an accompanying person. The analysis of 3 or 4 samples was performed for staff or hospitalized patients without COVID-19 symptoms.

When the sample tested negative, we excluded the entire population subjected to the test thanks to that single analysis; when the sample tested positive, the search for the positive subject was carried out by dividing the population and reanalyzing them individually. This strategy has already been demonstrated and widely successfully adopted in a number of studies performed with another method [5,6,7,8,9].

## 3. Results

All the positive swabs analyzed using VitaPCR platform (Menarini Diagnostics and Credo Diagnostics Biomedical) were then confirmed with RT-PCR analysis performed at SARS-CoV-2 regional reference centers.

Figure 2 shows the real-time PCR results of an asymptomatic subject who presented a positive SARS-CoV-2 swab that was firstly analyzed individually (Figure 1a) and then combined with a negative sample (Figure 1b). The results of the mean (±SD) of the two Ct values show the following values: SARS-CoV-2 specific RNA (yellow): mean 34.25 (±0.35), SARS-like universal RNA (red): mean 34.75 (±1.06).

The pooling test was also performed in a patient with febrile symptoms, analyzed individually and then together with a SARS-CoV-2 negative patient with the following results: mean SARS-CoV-2 = 19.5 (±0.71) and SARS-like universal RNA = 22 (±1.2) (Figure 3). The pooling test was tested on 20 SARS-CoV-2 positive patients. These samples were initially tested individually and then combined with a negative sample. In all the analyzed cases, the SD was analyzed starting from Ct values. For all cases, these Ct values remained within the limits for both SARS-CoV-2 specific RNA (yellow) and SARS-like universal RNA (red): min ± 0.35 and max ± 1.4. 

Of the 2251 swab tests screened, 52 (2.31%) were positive while 2199 (97.69%) were negative. Thanks to the use of three VitaPCR devices and the pooled analysis, it was possible to analyze a high number of subjects daily (from 4 to 123 subjects/day—mean of analyzed samples 47.9; SD ± 25.4). In particular, 292 subjects were analyzed in a pool of four samples (only one pool tested positive, 1.37%), 768 subjects were analyzed in a pool of three samples (six pools tested positive, 2.34%); 776 subjects were analyzed in a pool of two samples (six positive pools, 1.5%), and 415 subjects were analyzed individually (39 positive results) (Table 1). The standard deviation of Ct values in the pools with different sample sizes (from two to four) remained for seven cases analyzed within the limits (min ± 0.3 and max ± 1.1) for both SARS-CoV-2 specific RNA and SARS-like universal RNA. The pooling of five or more samples together was not tested in order to avoid a long waiting time for subjects performing the swab test for the retesting in case of positivity. In summary, of the 717 swabs (pooling) carried out, only 13 swabs (1.8%) were positive. The oropharyngeal/nasopharyngeal swab was then repeated for all 34 subjects (13 pools), and only 18 (53%) tested positive for SARS-CoV-2. A total of 1132 kits were used to analyze all 2251 subjects, representing a saving of ~50% in consumables/equipment, technical staff, and time.

All subjects who tested positive for SARS-CoV-2—confirmed when reanalyzed at the regional reference center—demonstrated the reliability and sensitivity of the VitaPCR SARS-CoV-2 test in detecting positive subjects. The single swab, however, has the limit of capturing the patient’s state in a very precise moment. In fact, the test may not detect positivity in the incubation period. The VitaPCR SARS-CoV-2 test detects 3.0 copies/μL of RNA (cDNA) transcripts.

## 4. Discussion

Our results show that over a range of pool sizes (2–4 individuals), Ct value differences between pooled tests with a positive sample and single individual positive samples remain almost unchanged. Particularly, the pooling test tested on 20 SARS-CoV-2 positive patients showed, in all cases, that SD of Ct value remained within the ranges for both SARS-CoV-2 specific RNA (yellow) and SARS-like universal RNA (red): min ± 0.35 and max ± 1.4. This variation of the Ct value detected, even in the absence of dilution effects, can be explained by the fact that it is a rapid semiquantitative RT-PCR test. To our knowledge, this is the first study reporting the pooling test applied in point-of-care testing thanks to a semiquantitative instrument. Similar studies have already detailed the qPCR sample pooling method [8,9]. In these studies, unlike ours, the pooled samples are diluted, resulting in less viral genetic material available to detect and thus in a greater likelihood of false-negative results. The novelty of our study is the lack of dilution of the analyzed pool; indeed, the same reaction buffer is used to analyze the oropharyngeal swab of the pooled subjects. There are no false-negative results even when the patient viral load is low or in weak positive samples, within the limits of 3.0 copies/μL of RNA (cDNA) transcripts of SARS-CoV-2.

Moreover, we have been using the pooling strategy (aggregation of up to four swabs in a test) to periodically monitor Oasi’s personnel. The results obtained from the swabs analyzed were confirmed a posteriori by checking the possible presence of symptoms over time and the presence of any outbreaks of infection within 47 working days. A limitation of pooling could concern the minor/absent quantity of cells taken from one of the oropharyngeal/nasopharyngeal samples. This limitation could be mitigated by requiring the swab to be performed by specialized healthcare staff and by using an appropriate collection procedure [10]. The pooling procedure leads, however, to numerous advantages. In fact, it is not necessary to increase the number of equipment and consumables, and additional staff training is not required. The pooling method is compatible with the VitaPCR SARS-CoV-2 platform (Menarini Diagnostics and Credo Diagnostics Biomedical) or similar equipment. This procedure provides almost immediate results (20 min), as in the event of a positive pool analysis, a prompt analysis by single subject is carried out. Unfortunately, pooled analysis cannot always be applied in population screening for COVID-19 as the use of single patient analysis is required very often. This is the case, for example, of already symptomatic subjects or subjects who have come into contact with positive subjects. It is a very advantageous procedure when it is necessary to periodically test the healthcare staff or the long-term patients.

## 5. Conclusions

The strategy implemented in our study has allowed us to significantly reduce (from 50 to 75%) times for analysis, equipment, consumables, and personnel costs while maintaining the same sensitivity. In addition, the possibility to be admitted to the hospital on the same day on which the swab is performed is extremely important for our patients (mainly affected by intellectual disability/dementia). Given the current epidemiological situation with a significant increase in PCR tests, such measures could help laboratories to lighten their workload and contain costs.

## Figures and Tables

**Figure 1 ijerph-18-09317-f001:**
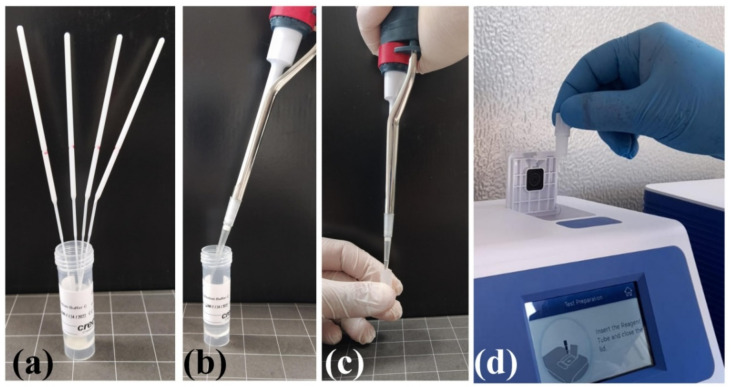
Two to four biological samples (oropharyngeal and nasopharyngeal swabs) are put in the same reaction buffer in the absence of dilution effects (**a**). Thirty microliters of the buffer (**b**) is placed in a lyophilized reagent (**c**) and loaded into the VitaPCR platform (**d**).

**Figure 2 ijerph-18-09317-f002:**
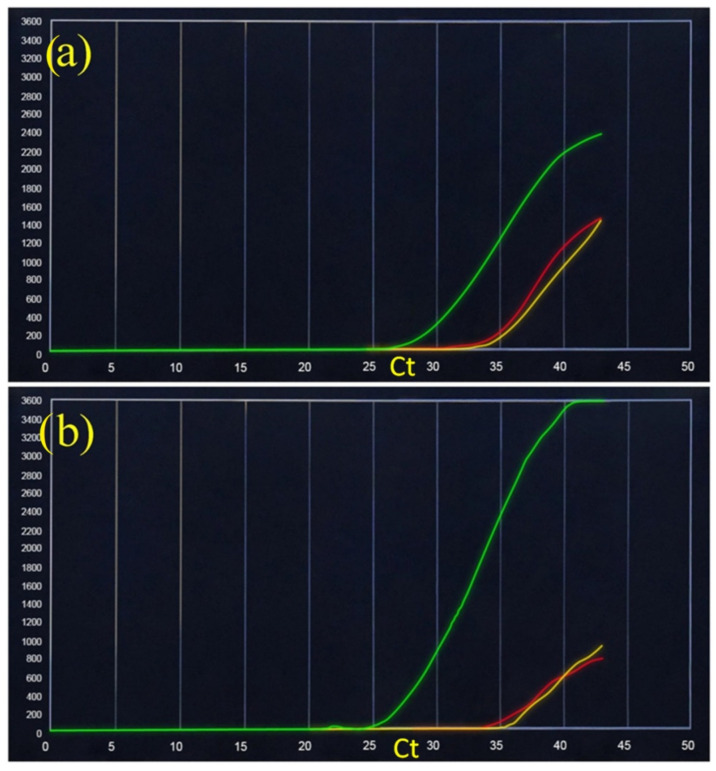
SARS-CoV-2 rapid RT-PCR-based diagnostic assay performed on the VitaPCR platform. The three fluorescent channels, FAM, VIC, and ROX, detect SARS-CoV-2 specific RNA (yellow), SARS-like universal RNA (red), and sample adequacy control (SAC) human beta-globin (green), respectively. (**a**) Positive asymptomatic SARS-CoV-2 patient; (**b**) positive SARS-CoV-2 patient pooled with a negative control.

**Figure 3 ijerph-18-09317-f003:**
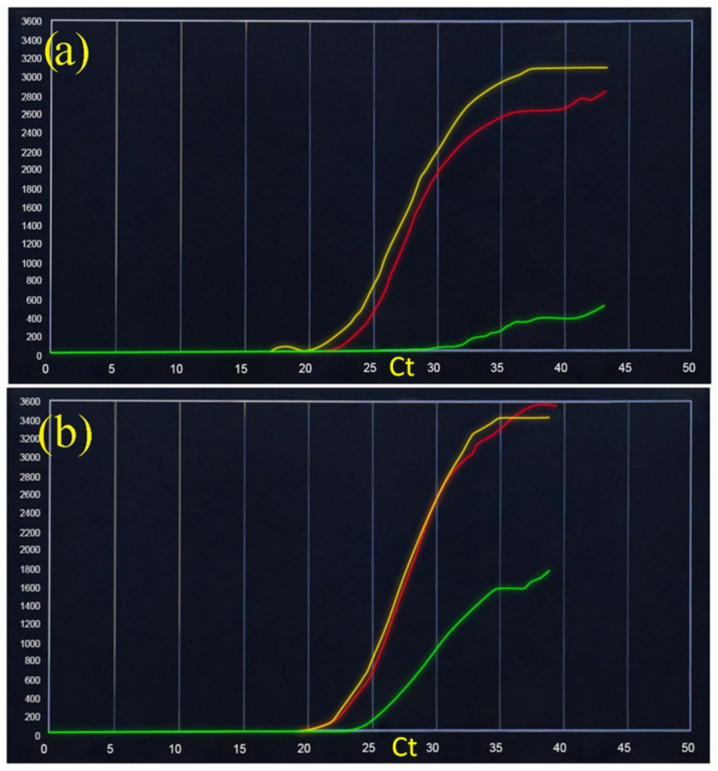
SARS-CoV-2 rapid RT-PCR-based diagnostic assay performed on the VitaPCR platform. The three fluorescent channels, FAM, VIC, and ROX, detect SARS-CoV-2 specific RNA (yellow), SARS-like universal RNA (red), and sample adequacy control (SAC) human beta-globin (green), respectively. Positive with febrile symptoms SARS-CoV-2 patient (**a**) and pooled with a negative control (**b**).

**Table 1 ijerph-18-09317-t001:** Sample pooling procedure using rapid SARS-CoV-2 diagnostic real-time PCR.

Pool	Subjects Analyzed	Sample Pooling	Positive for SARS-CoV-2
4	292	73	1
3	768	256	6
2	776	388	6
No pooling	415		39
	Total = 2251	Total = 717	Total = 52

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
