# Peer review of "Implementation of Sample Pooling Procedure Using a Rapid SARS-CoV-2 Diagnostic Real-Time PCR Test Performed Prior to Hospital Admission of People with Intellectual Disabilities"

_ijerph, 2021, doi:10.3390/ijerph18179317_

Round 1

Reviewer 1 Report

This is an original research that validates a sample pooling technique for the diagnosis/screening of SARS-CoV-2 infection. The research is well-designed, and the methods are clearly described. In the discussion, the authors also state the advantages of this procedure in comparison to other protocols described in the literature. The use of English is also adequate, and it is easy to go through the manuscript. I should recommend writing down the units (i.e. Ct) in each axis of the graphics, for a better understanding of the figures. Otherwise, I believe it is suitable for publication.  

Author Response

Dear Reviewers,

I would like to thank you for your valued comments and suggestions to the article. As you requested, we made all the necessary changes in our manuscript to address your concerns and we detailed below how the points raised have been accommodated. The main changes are highlighted in yellow in the text of the manuscript. From the changes made in the revised manuscript and responses provided below, I hope you are convinced that we have adequately addressed the reviewer’s concerns and made the paper better. If there are any further questions, please feel free to let me know.

Sincerely,

Luigi Vetri,

Reviewer Comments:

Reviewer 1

This is an original research that validates a sample pooling technique for the diagnosis/screening of SARS-CoV-2 infection. The research is well-designed, and the methods are clearly described. In the discussion, the authors also state the advantages of this procedure in comparison to other protocols described in the literature. The use of English is also adequate, and it is easy to go through the manuscript. I should recommend writing down the units (i.e. Ct) in each axis of the graphics, for a better understanding of the figures. Otherwise, I believe it is suitable for publication.  

Many thanks for you valued suggestions. We corrected the units in each axis as you requested.

Reviewer 2 Report

The authors have addressed my previous minor comments and suggestions.

Author Response

Dear Reviewers,

I would like to thank you for your valued comments and suggestions to the article. As you requested, we made all the necessary changes in our manuscript to address your concerns and we detailed below how the points raised have been accommodated. The main changes are highlighted in yellow in the text of the manuscript. From the changes made in the revised manuscript and responses provided below, I hope you are convinced that we have adequately addressed the reviewer’s concerns and made the paper better. If there are any further questions, please feel free to let me know.

Sincerely,

Luigi Vetri

Reviewer 2

The authors have addressed my previous minor comments and suggestions.

Many thanks for you valued suggestions.

Reviewer 3 Report

In the review of the manuscripts, it appears that the authors have tried to improve the quality of the manuscript, which is appreciated. However the presentation of amplification curves is incomplete. The CT value averages/ the calculations of relative expressions are lacking. 

It is difficult to understand the methods and the interpretation of results is made difficult.

Given that pooling tests for SARS-CoV2 are already accepted and established, how is this procedure novel?

Author Response

Dear Reviewers,

I would like to thank you for your valued comments and suggestions to the article. As you requested, we made all the necessary changes in our manuscript to address your concerns and we detailed below how the points raised have been accommodated. The main changes are highlighted in yellow in the text of the manuscript. From the changes made in the revised manuscript and responses provided below, I hope you are convinced that we have adequately addressed the reviewer’s concerns and made the paper better. If there are any further questions, please feel free to let me know.

Sincerely,

Luigi Vetri

Reviewer 3

In the review of the manuscripts, it appears that the authors have tried to improve the quality of the manuscript, which is appreciated. However the presentation of amplification curves is incomplete.

Many thanks for the suggestsion. We rewrote the units (Ct) in each axis of the graphics, for a better understanding of the figures.

The CT value averages/ the calculations of relative expressions are lacking.

We changed “average” with “mean” in the Materials and Methods paragraph. The mean of the Ct values and the relative standard deviation (SD) are indicated in the result paragraph.

 It is difficult to understand the methods and the interpretation of results is made difficult. Given that pooling tests for SARS-CoV2 are already accepted and established, how is this procedure novel?

Thanks for the suggestion. The main novelty of our study is that in other similar studies [8-9] the pooled samples are diluted, resulting in less viral genetic material available to detect, and then with a greater likelihood of false-negative results. The novelty of our study is the lack of dilution of the analyzed pool; indeed, the same reaction buffer is used to analyze the oro-pharyngeal swab of the pooled subjects. Please see lines 191-199.

This manuscript is a resubmission of an earlier submission. The following is a list of the peer review reports and author responses from that submission.

Round 1

Reviewer 1 Report

The work focuses on reducing the cost of time and efforts in detecting SARS-CoV-2 infections using the 2251 swabs samples collected from patients with intellectual disabilities. Their study shows that by using a pooling method, the cost of the time, stuff and equipment was reduced by nearly 50%, while the sensitivity and reliability of the detection remained high. The manuscript is overall in a good shape, but I do have the following suggestions before the paper can be considered for publication. 

To improve:

  1. Please explain the pooling strategy on why specific samples are chosen for pools 2, 3, 4 respectively.
  2. What is the comparison on the reliability and sensitivity in the pools with different sample size (from 2 to 4)? What about pooling 5 or more sample together? Is the size of 4 at the limit of the sample sizes? Please address with more analytical data.
  3. Please address the significance and novelty of this work when compared to other similar studies.

minor error: 

  1. In the result section, please correct "SARS-CoV = 22 (± 1.2)”
  2. In the result section, "of the 716 swabs carried out", please specify the pooled swabs, and it should be in total 717 swabs pools. 

Reviewer 2 Report

Musumeci and cols. shown a brief report manuscript describing the implementation of sample pooling procedure using a rapid SARS-CoV-2 Diagnostic RT PCR test preceding to hospital admission of people with intellectual disabilities.

Without a doubt, the topic is extremely attractive in the course of the current pandemic. Therefore, the authors test the sample pooling (up to 4 samples) in a single test.

In general, the research looks well-designed, and the paper is written politely.

I just have some minor comments:

In the results section, the paragraph describing the 2251 swab tests screened would be better in a small table.

The included figure should be improved in quality.

Please be sure to homogenize terms, for example, SARS-CoV-2 and COVID-19.

Reviewer 3 Report

The current manuscript, 'Implementation of sample pooling procedure using a rapid SARS-CoV-2 Diagnostic Real-Time PCR test performed prior  hospital admission of people with intellectual disabilities', tested the possibility of pooling samples to perform COVID 19 tests by RT-PCR and concluded that this procedure can be valuable and reliable. The major concern with the paper is that:

  1. There is a complete lack of novelty in this manuscript, as a number of manuscripts have shown this result before: Mahmoud et al., 2021;  Sawiki et al., 2021; just to mention a few. These studies are all more detailed and thorough characterization of the qPCR sample pooling method. The only difference here is that it was done in individuals with intellectual disabilities, which is a variable that doesn't influence the outcome.
  2. The explanation of methods and detailing of results and discussion is extremely inadequate. Moreover, only a model amplification curve without actual data analysis/ statistical methods is not acceptable.
  3. FDA had approved the usage of this technique in June 2020, almost a year ago: https://www.fda.gov/medical-devices/coronavirus-covid-19-and-medical-devices/pooled-sample-testing-and-screening-testing-covid-19.  Since that was based on extensive evidence, another manuscript considering the same system with nothing new, adds nothing more to the field.